# The structural connectome constrains fast brain dynamics

**Pierpaolo Sorrentino**[1,2,3,4]*, **Caio Seguin**[5], **Rosaria Rucco**[2,3], **Marianna Liparoti**[2,3], **Emahnuel Troisi Lopez**[2,3], **Simona Bonavita**[6], **Mario Quarantelli**[7], **Giuseppe Sorrentino**[4], **Viktor Jirsa**[1†], **Andrew Zalesky**[5†]

[1]Aix-Marseille University, Inserm, INS, Institut de Neurosciences des Systèmes, Marseille, France; [2]Department of Motor Sciences and Wellness, Parthenope University of Naples, Naples, Italy; [3]Institute for Diagnosis and Cure Hermitage Capodimonte, Naples, Italy; [4]Institute of Applied Sciences and Intelligent Systems, National Research Council, Pozzuoli, Italy; [5]University of Melbourne, Melbourne, Australia; [6]University of Campania Luigi Vanvitelli, Caserta, Italy; [7]Biostructure and Bioimaging Institute, National Research Council, Naples, Italy

**Abstract** Brain activity during rest displays complex, rapidly evolving patterns in space and time. Structural connections comprising the human connectome are hypothesized to impose constraints on the dynamics of this activity. Here, we use magnetoencephalography (MEG) to quantify the extent to which fast neural dynamics in the human brain are constrained by structural connections inferred from diffusion MRI tractography. We characterize the spatio-temporal unfolding of whole-brain activity at the millisecond scale from source-reconstructed MEG data, estimating the probability that any two brain regions will significantly deviate from baseline activity in consecutive time epochs. We find that the structural connectome relates to, and likely affects, the rapid spreading of neuronal avalanches, evidenced by a significant association between these transition probabilities and structural connectivity strengths (r = 0.37, p<0.0001). This finding opens new avenues to study the relationship between brain structure and neural dynamics.

*For correspondence:
pierpaolo.SORRENTINO@univ-amu.fr

†These authors contributed equally to this work

**Competing interests:** The authors declare that no competing interests exist.

## Introduction

The structural scaffolding of the human connectome (*Sporns et al., 2005*) constrains the unfolding of large-scale coordinated neural activity towards a restricted *functional repertoire* (*Deco et al., 2011*). While functional magnetic resonance imaging (fMRI) can elucidate this phenomenon at relatively slow timescales (*Honey et al., 2007*; *Goñi et al., 2014*; *Zalesky et al., 2014*), brain activity shows rich dynamic behaviour across multiple time scales, with faster activity nested within slower scales. Here, in healthy young adults, we exploit the high temporal resolution of resting-state magnetoencephalography (MEG) data to study the spatial spread of perturbations of local activations representative of neuronal avalanches. We aim to establish whether the structural connectome constrains the spread of avalanches among regions (*Beggs and Plenz, 2004*; *Shriki et al., 2013*). We find that avalanche spread is significantly more likely between pairs of grey matter regions that are structurally connected, as inferred from diffusion MRI tractography. This result provides cross-modal empirical evidence suggesting that connectome topology constrains fast-scale transmission of neural information, linking brain structure to brain dynamics.

## Results

Structural connectomes were mapped for 58 healthy adults (26 females, mean age ± SD: 30.72 ± 11.58) using diffusion MRI tractography and regions defined based on the Automated Anatomical

Labeling (AAL) and the Desikan–Killiany–Tourville (DKT) atlases. Interregional streamline counts derived from whole-brain deterministic tractography quantified the strength of structural connectivity between pairs of regions. Streamline counts were normalized by regional volume. Group-level connectomes were computed by averaging connectivity matrices across participants.

MEG signals were pre-processed and source reconstructed for both the AAL and DKT atlases. All analyses were conducted on source-reconstructed signal amplitudes. Each signal amplitude was z-scored and binarized such that, at any time point, a z-score exceeding a given threshold was set to 1 (active); all other time points were set to 0 (inactive). An avalanche was defined as starting when any region exceeded this threshold, and finished when no region was active. An avalanche-specific transition matrix (TM) was calculated, where element $(i, j)$ represented the probability that region $j$ was active at time $t + \delta$, given that region $i$ was active at time $t$, where $\delta \sim 3$ ms. The TMs were averaged per participant, and then per group, and finally symmetrized. *Figure 1* provides an overview of the pipeline.

We found striking evidence of an association between avalanche transition probabilities and structural connectivity strengths (*Figure 2*), suggesting that regional propagation of fast-scale neural avalanches is partly shaped by the axonal fibres forming the structural connectome (r = 0.40, p<0.0001). Specifically, the association was evident for different activation thresholds and both the AAL and DKT connectomes (AAL atlas: for threshold z = 2.5, r = 0.41; for threshold z = 3.0, r = 0.40; for threshold z = 3.5, r = 0.39; DKT atlas: for threshold z = 2.5, r = 0.38; for threshold z = 3.0, r = 0.37; for threshold z = 3.5, r = 0.35; in all cases, p<0.0001), as well as for individual- and group-level connectomes, although associations were stronger for group-level analyses (see *Figure 2A*).

We also investigated this phenomenon within specific frequency bands. Associations were evident in all the classical frequency bands: delta (0.5–4 Hz; r = 0.39), theta (4–8 Hz; r = 0.29), alpha (8–13 Hz; r = 0.32), beta (13–30 Hz; r = 0.32), and gamma (30–48 Hz; r = 0.32), with p<0.0001 for all bands (see *Supplementary file 1*). Supplementary analyses suggested that these results could not be attributable to volume conduction confounds (see section Field spread analysis).

Next, we sought to test whether the associations were weaker for randomized TMs computed after randomizing the times of each avalanche while keeping the spatial structure unchanged. Randomized TMs resulted in markedly weaker associations with structural connectivity compared to the actual TMs (AAL atlas, z-score = 3: mean r = 0.26, observed r = 0.40, p<0.001). Note that the mean correlation coefficient was greater than zeros for the randomized data because the randomization process preserved basic spatial attributes in the data. We also found that the findings remained significant after excluding subcortical regions (with lower signal-to-noise ratios). Finally, we replicated these findings for a group-level connectome derived using diffusion MRI acquired from 200 healthy adults in the Human Connectome Project (r = 0.11, p<0.001, z-score = 3; see Materials and methods). Our results were thus robust to multiple connectome mapping pipelines and parcellation atlases, significant for both group-averaged and individual connectomes, and could not be explained by chance transitions and/or volume conduction effects. Collectively, these results suggest that connectome organization significantly shapes the propagation of neural activity.

## Discussion

Our results provide new insight into the propagation of fast-evolving brain activity in the human connectome. We show that the spatial unfolding of neural dynamics at the millisecond scale relates to the network of large-scale axonal projections comprising the connectome, likely constraining the exploration of the brain's putative functional repertoire. The short time scale of several milliseconds biases the constraint to direct connections, which is the focus of this paper. Longer delays may impose constraints upon larger-scale motifs of the network and further characterize the sub-spaces, in which brain dynamics unfold.

Previous functional MRI studies provide evidence of coupling between structural connectivity and slow activations (*Honey et al., 2007*; *Honey et al., 2010*; *Honey et al., 2009*). However, intrinsic neural dynamics evolve quickly and are nested within slow activity (*Saggio et al., 2017*). Our findings suggest that long-term structure-function coupling occurs against a backdrop of faster fluctuations, which are also constrained by the connectome and may enable individuals to rapidly respond to changing environments and new cognitive demands (*McIntosh and Jirsa, 2019*).

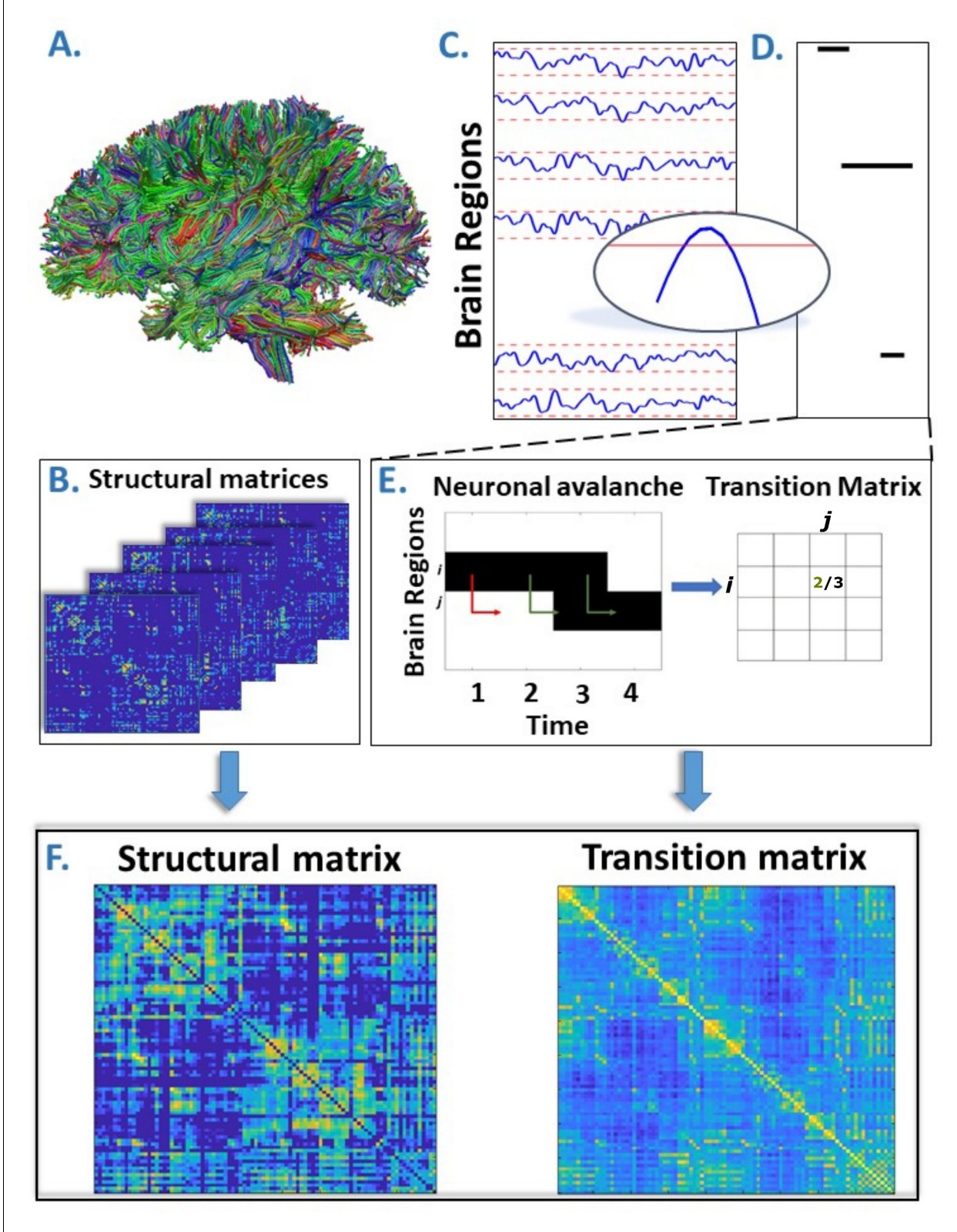

**Figure 1.** Overview of the pipeline. (**A**) Rendering of streamlines reconstructed using diffusion magnetic resonance imaging and tractography for an individual. (**B**) Structural connectivity matrix. Row/columns represent regions comprising a brain atlas. Matrix entries store the number of streamlines interconnecting each pair of regions. (**C**) Source-reconstructed magnetoencephalography series. Each blue line represents the z-scored activity of a region, and the red lines denote the threshold (z-score = ±3). The inset represents a magnified version of a time series exceeding the threshold. (**D**)

*Figure 1 continued on next page*

*Figure 1 continued*

Raster plot of an avalanche. For each region, the moments in time when the activity is above threshold are represented in black, while the other moments are indicated in white. The particular avalanche that is represented involved three regions. (E) Estimation of the transition matrix of a toy avalanche. Region $i$ is active three times during the avalanche. In two instances, denoted by the green arrows, region $j$ was active after region $i$. In one instance, denoted by the red arrow, region $i$ is active but region $j$ does not activate at the following time step. This situation would result, in the transition matrix, as a 2/3 probability. (F) Average structural matrix and average transition matrix (log scale).

Consistent with our findings, two recent M/EEG studies showed that functional connectivity, as estimated using amplitude-envelope coupling (AEC), relates to structural connectivity (*Glomb et al., 2020*; *Tewarie et al., 2019*). However, in contrast to AEC, we conducted time-resolved analyses, characterizing avalanche dynamics at high temporal resolution. Further work is needed to determine the extent to which structure-function coupling is dynamic. To this regard, our results suggest that coupling is strongest during avalanche events, consistent with established theories (*Dehaene et al., 1998*). Finally, our results might explain how the large-scale activity unfolding in time might lead to the previous observation that average resting-state functional connectivity displays topological features that mirror those of the structural connectome (*Bullmore and Sporns, 2009*). Our proposed framework links the large-scale spreading of aperiodic, locally generated perturbations to the structural connectome and might be further exploited to investigate polysynaptic models of network communication, which aim to describe patterns of signalling between anatomically unconnected regions (*Seguin et al., 2018*; *Seguin et al., 2019*). In fact, our results show that transitions of activations are observed across regions that do not appear to be directly linked in the structural connectome. This provides evidence for polysynaptic communication.

Neuronal avalanches have been previously observed in MEG data (*Shriki et al., 2013*), and their statistical properties, such as a size distribution that obeys a power-law with an exponent of $-3/2$, reported. These features are compatible with those that would be predicted starting from a process

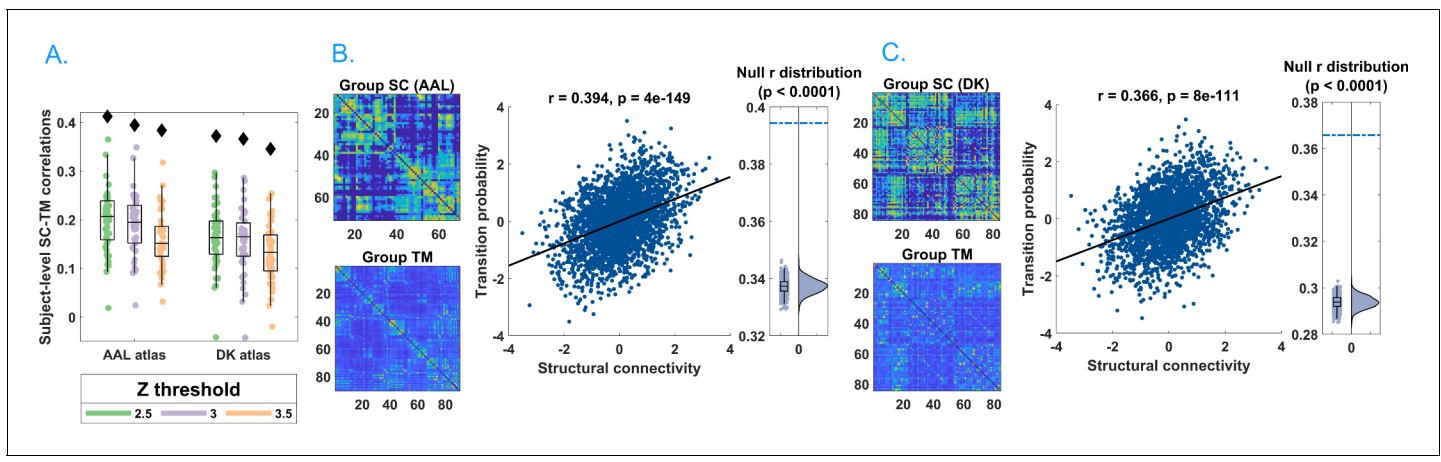

**Figure 2.** Main results. (A) Distribution of the r's of the Spearman's correlation between the subject-specific transition matrices and structural connectomes. The black diamond represent the r's of the group-averaged matrices. On the left, the results for the Automated Anatomical Labeling (AAL) atlas; on the right, the results for the Desikan–Killiany–Tourville (DKT) atlas. Green, purple, and orange dots represent results obtained with a z-score threshold of 2.5, 3, and 3.5, respectively. (B, C) Data referring to the AAL atlas in (B) and DKT atlas in (C). On the top left, the average structural matrix; on the bottom left, the average transition matrix. The scatterplot shows the correlation between the values of the structural edges and the transition probabilities for the corresponding edge. The black line represents the best fit line in the least-square sense. On the right, the distribution shows the r's derived from the null distribution. The dotted blue line represents the observed r. Please note that, for visualization purposes, the connectivity weights and the transition probabilities were resampled to normal distributions. *Figure 2—figure supplement 1* shows the comparison between the structural connectome and the transition matrix computed by taking into account longer delays. In *Supplementary file 1*, we report a table with an overview of the results of the frequency-specific analysis.

The online version of this article includes the following source data and figure supplement(s) for figure 2:

**Source data 1.** This source data file contains the code to generate the transition matrices starting from neuronal avalanches and to compare them to null surrogates.

**Figure supplement 1.** On the left, the average structural matrix.

operating at criticality with a branching ratio equal to 1. While beyond the scope of this paper, our framework might contribute to elucidating the role of the structural scaffolding (and its topological properties) to the emergence of the observed large-scale, scale-free critical dynamics. In turn, this might be exploited to predict the effects of structural lesions on behaviour and/or clinical phenotypes.

While our findings were replicated across multiple frequency bands, structural connectivity can potentially impose frequency-dependent constraints on avalanche spread. Future work should investigate frequency-specific data to understand what leads to the emergence of avalanches and, most importantly, to the specific spatio-temporal patterns of recruited regions that defines individual (or at least groups of) avalanches in each specific frequency band.

For the present application, we reconstructed the structural connectome using a deterministic tractography algorithm. While probabilistic algorithms can provide advantages in some applications, they are prone to reconstruction of spurious connections (false positives), compared to deterministic methods, reducing connectome specificity (*Sarwar et al., 2019*; *Zalesky et al., 2016*). We used deterministic tractography because previous functional MRI studies report that structure-functional coupling is greater for connectivity matrices inferred from deterministic tractography compared to probabilistic methods (*Abeyasinghe et al., 2021*). Nonetheless, additional studies are needed to clarify if and to what extent the present results are influenced by the structural connectome reconstruction method. While we replicated our findings using alternative datasets (i.e., Human Connectome Project [HCP]) and parcellations, further replication using alternative connectome mapping pipelines is warranted.

In conclusion, using MEG to study fast neuronal dynamics and diffusion MRI tractography to map connectomes, we found that the connectome significantly constrains the spatial spread of neuronal avalanches to axonal connections. Our results suggest that large-scale structure-function coupling is dynamic and peaks during avalanche events.

## Materials and methods

### Participants

We recruited 58 young adults (males 32/females 26, mean age ± SD was 30.72 ± 11.58) from the general community. All participants were right-handed and native Italian speakers. The inclusion criteria were (1) no major internal, neurological, or psychiatric illnesses; and (2) no use of drugs or medication that could interfere with MEG/MRI signals. The study complied with the Declaration of Helsinki and was approved by the local Ethics Committee. All participants gave written informed consent.

### MRI acquisition

3D T1-weighted brain volumes were acquired at 1.5 Tesla (Signa, GE Healthcare) using a 3D Magnetization-Prepared Gradient-Echo BRAVO sequence (TR/TE/TI 8.2/3.1/450 ms, voxel $1 \times 1 \times 1$ mm$^3$, 50% partition overlap, 324 sagittal slices covering the whole brain), and diffusion MRI data for individual connectome reconstruction were obtained using the following parameters: Echo-Planar Imaging, TR/TE 12,000/95.5 ms, voxel $0.94 \times 0.94 \times 2.5$ mm$^3$, 32 diffusion-sensitizing directions (5 B0 volumes). The MRI scan was performed after the MEG recording. Pre-processing of the diffusion MRI data was carried out using the software modules provided in the FMRIB Software Library (FSL, http://fsl.fmrib.ox.ac.uk/fsl). All diffusion MRI datasets were corrected for head movements and eddy currents distortions using the 'eddy_correct' routine, rotating diffusion sensitizing gradient directions accordingly, and a brain mask was obtained from the B0 images using the Brain Extraction Tool routine. A diffusion-tensor model was fitted at each voxel, and streamlines were generated over the whole brain by deterministic tractography using Diffusion Toolkit (FACT propagation algorithm, angle threshold 45°, spline-filtered, masking by the Fractional Anisotropy (FA) maps thresholded at 0.2). For tractographic analysis, the regions of interest (ROIs) of the AAL atlas and of an MNI space-defined volumetric version of the DKT ROI atlas were used, both masked by the gray matter (GM) tissue probability map available in Statistical Parametric Mapping Software - SPM (thresholded at 0.2). To this end, for each participant, FA volumes were normalized to the MNI space using the FA template provided by FSL, using the spatial normalization routine available in

SPM12, and the resulting normalization matrices were inverted and applied to the ROIs, to apply them onto each subject. The quality of the normalization was assessed visually. From each subject's whole-brain tractography and corresponding GM ROI set, the number of streamlines connecting each couple of GM ROIs and the corresponding mean tract length was calculated using an in-house software written in Interactive Data Language (IDL, Harris Geospatial Solutions, Inc, Broomfield, CO).

Connectomes in the replication dataset were constructed using an alternative mapping pipeline and diffusion MRI data from the HCP. Deterministic tractography was performed using MRtrix3 (*Tournier et al., 2019*) under the following parameters: FACT algorithm, 5 million streamlines, 0.5 mm propagation step size, 400 mm maximum propagation length, and 0.1 FA threshold for the termination of streamlines (*Seguin et al., 2019*). The number of streamlines connecting any couple of regions was normalized by the combined volume of the two regions. Structural matrices were constructed for 200 HCP participants using the AAL atlas and averaged to derive a group-level connectome.

## MEG pre-processing

MEG pre-processing and source reconstruction were performed as in *Sorrentino et al., 2021*. The MEG system was equipped with 163 magnetometers and was developed by the National Research Council of Italy at the Institute of Applied Sciences and Intelligent Systems (ISASI). All technical details regarding the MEG device are reported in *Rombetto et al., 2014*. In short, the MEG registration was divided into two eyes-closed segments of 3:30 min each. To identify the position of the head, four anatomical points and four position coils were digitized. Electrocardiogram (ECG) and electro-oculogram (EOG) signals were also recorded. The MEG signals, after an anti-aliasing filter, were acquired at 1024 Hz, then a fourth-order Butterworth IIR band-pass filter in the 0.5–48 Hz band was applied. To remove environmental noise, measured by reference magnetometers, we used principal component analysis. We adopted supervised independent component analysis to clean the data from physiological artefacts, such as eye blinking (if present) and heart activity (generally one component). Noisy channels were identified and removed manually by an expert rater (136 ± 4 sensors were kept). 47 subjects were selected for further analysis.

## Source reconstruction

The time series of neuronal activity were reconstructed in 116 ROIs based on the AAL atlas (*Tzourio-Mazoyer et al., 2002*; *Hillebrand et al., 2016*); and in 84 regions of interest based on the DKT atlas. To do this, we used the volume conduction model proposed by *Nolte, 2003* applying the linearly constrained minimum variance (LCMV) beamformer algorithm (*Van Veen et al., 1997*) based on the native structural MRIs. Sources were reconstructed for the centroid of each ROI. Finally, we considered a total of 90 ROIs for the AAL atlas since we have excluded 26 ROIs corresponding to the cerebellum because of their low reliability in MEG (*Lardone et al., 2018*). All the pre-processing steps and the source reconstruction were made using the Fieldtrip toolbox (*Oostenveld et al., 2011*).

## Neuronal avalanches and branching parameter

To study the dynamics of brain activity, we estimated 'neuronal avalanches'. Firstly, the time series of each ROI was discretized calculating the z-score, then positive and negative excursions beyond a threshold were identified. The value of the threshold was set to three standard deviations ($|z| = 3$), but we tested the robustness of the results changing this threshold from 2.5 to 3.5. A neuronal avalanche begins when, in a sequence of contiguous time bins, at least one ROI is active ($|z| > 3$) and ends when all ROIs are inactive (*Beggs and Plenz, 2003*; *Shriki et al., 2013*). The total number of active ROIs in an avalanche corresponds to its size.

These analyses require the time series to be binned. This is done to ensure that one is capturing critical dynamics, if present. To estimate the suitable time bin length, for each subject, each neuronal avalanches, and each time bin duration, the branching parameter σ was estimated (*Haldeman and Beggs, 2005*; *Harris, 1964*). In fact, systems operating at criticality typically display a branching ratio ~1. The branching ratio is calculated as the geometrically averaged (over all the time bins) ratio of the number of events (activations) between the subsequent time bin (descendants) and that in the

current time bin (ancestors) and then averaging it over all the avalanches (*Bak et al., 1987*). More specifically:

$$\sigma_i = \frac{1}{N_{bin}-1} \prod_{j=1}^{N_{bin}-1} \left( \frac{n_{events}(j+1)}{n_{events}(j)} \right)^{\frac{1}{N_{bin}-1}} \tag{1}$$

$$\sigma = \frac{1}{N_{aval}} \prod_{i=1}^{N_{aval}} (\sigma_i)^{\frac{1}{N_{aval}}} \tag{2}$$

where $\sigma_i$ is the branching parameter of the ith avalanche in the dataset, $N_{bin}$ is the total amount of bins in the ith avalanche, $n_{events}(j)$ is the total number of events active in the jth bin, and $N_{aval}$ is the total number of avalanche in the dataset. We tested bins from 1 to 5, and picked 3 for further analyses, given that the branching ratio was 1 for bin = 3. However, results are unchanged for other bin durations, and the branching ratio remains equal to 1 or differences were minimal (range: 0.999–1.010 – data not shown). Bins of longer duration would violate the Nyquist criterion and were thus not considered. The results shown are derived when taking into account avalanches longer than 10 time bins. However, we repeated the analysis taking into account avalanches longer than 30 time bins, as well as taking all avalanches into account, and the results were unchanged.

## Transition matrices

The amplitude of each binned, z-scored source-reconstructed signal was binarized, such that, at any time bin, a z-score exceeding ±3 was set to 1 (active); all other time bins were set to 0 (inactive). Alternative z-score thresholds (i.e., 2.5 and 3.5) were tested. An avalanche was defined as starting when any region is above threshold and finishing when no region is active, as in *Sorrentino et al., 2021*. Avalanches shorter than 10 time bins (~30 ms) were excluded. However, the analyses were repeated including only avalanches longer than 30 time bins (~90 ms) to focus on rarer events (sizes of the neuronal avalanches have a fat-tailed distribution) that are highly unlikely to be noise, and including all avalanches, and the results were unchanged. An avalanche-specific TM was calculated, where element (*i*, *j*) represented the probability that region *j* was active at time $t + \delta$, given that region *i* was active at time *t*, where $\delta \sim 3$ ms. The TMs were averaged per participant, and then per group, and finally symmetrized. The introduction of a time lag makes it unlikely that our results can be explained trivially by volume conduction (i.e., the fact that multiple sources are detected simultaneously by multiple sensors, generating spurious zero lag correlations in the recorded signals). For instance, for a binning of 3, as the avalanches proceed in time, the successive regions that are recruited do so after roughly 3 ms (and 5 ms for the binning of 5). Hence, activations occurring simultaneously do not contribute to the estimate of the TM. See below for further analyses addressing the volume conduction issue. Finally, we explored TMs estimated using frequency-specific signals. To this end, we filtered the source-reconstructed signal in the classical frequency bands (delta, 0.5–4 Hz; theta 4–8 Hz; alpha 8–13 Hz; beta 13–30 Hz; gamma 30–48 Hz), before computing neuronal avalanches and the TM, by applying a fourth-order Butterworth pass-band filter to the source-reconstructed data, before proceeding to the further analysis as previously described. The results remained significant in all the explored frequency bands. This analysis was carried out for the DKT atlas, binning = 3, z-score threshold = ± 3.

## Field spread analysis

Volume conduction alone is an unlikely explanation of our results, given that simultaneous activations do not contribute to the TM, due to the time lags introduced. To confirm that volume conduction effects were negligible, the TMs were re-computed using longer delays. In short, we identified the regions that were recruited in an avalanche after the first perturbation (i.e., the initial time bin of an avalanche). Since we did not scroll through the avalanche in time, as previously described, we considered time delays as long as the avalanche itself, while minimizing the influence of short delays. This means that the avalanche-specific TM is now binary, and the *ij*th element is equal to 1 if region *i* started the avalanche (i.e., it was active at the first time bin) and region *j* was recruited in the avalanche at any subsequent time point, and 0 otherwise. This alternative procedure for the estimation of the TMs was carried out for the AAL atlas, in the case of binning = 3, z-score threshold = ± 3. In

this case, a significant association remained between transition probabilities and structural connectivity (r = 0.36; p<0.0001). *Figure 2—figure supplement 1* provides further details.

To further rule out the possibility that field spread might introduce spurious correlations that might drive the relationship between the TM and the structural connectivity matrix, we conducted further analyses involving surrogate data. We generated n white Gaussian processes, with n = 66, that is, the number of cortical regions, and we smoothed them using a zero-phase polynomial filter. Then, we added 100 perturbations, where each perturbation was assigned to a randomly chosen regions and random time point, subject to the following constraints. Perturbations were separated by at least 200 samples (no overlap was allowed, i.e., the perturbations could only occur in one region at a time), their length was randomly selected among 5, 10, or 100 samples, their amplitude between 50 and 400. This procedure was carried out 47 times to obtain an independent surrogate dataset for each one of the 47 participants, which will be referred to as the 'uncoupled' dataset. The uncoupled dataset was then transformed using the subject-specific leadfield matrix, yielding new surrogate sensor-level time series, where each sensor is a weighted sum of all the sources, according to the same leadfield matrix that was used to reconstruct the real data. Noise, correlated as 1/*distance among sensors*, was then added to the sensor-level time series, with an SNR = 4. Then, new source-reconstructed time series were computed for each subject. Based on these new time series, we performed the same procedure to compute the TM as described above. Specifically, we z-scored the time series, thresholded them (threshold z = ±3), retrieved the avalanche-specific TMs, averaged these within each subject and then across the group, and finally symmetrized the matrix. We then investigated the extent of the correlation between the new TM and the structural connectivity matrix. We repeated the entire procedure reported above 100 times and show that is unlikely that linear mixing alone can explain the significant association between transition probabilities and structural connectivity (p<0.001).

## Statistical analysis

The Spearman rank correlation coefficient was used to assess the association between transition probabilities and structural connectivity. A correlation coefficient was computed separately for each individual across all pairs of regions. TMs were symmetrized before this computation. Randomized TMs were generated to ensure that associations between transition probabilities and structural connectivity could not be attributed to chance. Avalanches were randomized across time, without changing the order of active regions at each time step. We generated a total of 1000 randomized TMs and the Spearman rank correlation coefficient was computed between each randomized matrix and structural connectivity. This yielded a distribution of correlation coefficients under randomization. The proportion of correlation coefficients that were greater than, or equal to, the observed correlation coefficient provided a p-value for the null hypothesis that structure-function coupling was attributable to random transition events.

## Additional information

### Funding
No external funding was received for this work.

### Author contributions
Pierpaolo Sorrentino, Conceptualization, Data curation, Formal analysis, Validation, Investigation, Visualization, Methodology, Writing - original draft, Writing - review and editing; Caio Seguin, Rosaria Rucco, Conceptualization, Data curation, Formal analysis, Methodology, Writing - review and editing; Marianna Liparoti, Emahnuel Troisi Lopez, Methodology; Simona Bonavita, Writing - review and editing; Mario Quarantelli, Data curation, Methodology; Giuseppe Sorrentino, Resources, Data curation, Funding acquisition, Methodology, Writing - review and editing; Viktor Jirsa, Conceptualization, Formal analysis, Methodology, Writing - original draft, Writing - review and editing; Andrew Zalesky, Formal analysis, Supervision, Validation, Visualization, Writing - original draft, Writing - review and editing

## Author ORCIDs
Pierpaolo Sorrentino (iD) https://orcid.org/0000-0002-9556-9800
Rosaria Rucco (iD) http://orcid.org/0000-0003-0943-131X
Marianna Liparoti (iD) http://orcid.org/0000-0003-2192-6841
Emahnuel Troisi Lopez (iD) http://orcid.org/0000-0002-0220-2672
Mario Quarantelli (iD) https://orcid.org/0000-0001-7836-454X
Giuseppe Sorrentino (iD) http://orcid.org/0000-0003-0800-2433
Viktor Jirsa (iD) https://orcid.org/0000-0002-8251-8860

## Ethics

Human subjects: All participants gave written informed consent. The study complied with the declaration of Helsinki and was approved by the local Ethics Committee (Prot.n.93C.E./Reg. n.14-17OSS).

## Decision letter and Author response

Decision letter https://doi.org/10.7554/eLife.67400.sa1
Author response https://doi.org/10.7554/eLife.67400.sa2

## Additional files

### Supplementary files
• Supplementary file 1. Correlations between the structural connectome and frequency-specific transition matrices.
• Transparent reporting form

### Data availability

The MEG data and the reconstructed avalanches are available upon request to the corresponding author (Pierpaolo Sorrentino), conditional on appropriate ethics approval at the local site. The availability of the data was not previously included in the ethical approval, and therefore data cannot be shared directly. In case data are requested, the corresponding author will request an amendment to the local ethical committee. Conditional to approval, the data will be made available. The Matlab code is available at https://github.com/pierpaolosorrentino/Transition-Matrices- (copy archived at https://archive.softwareheritage.org/swh:1:rev:3846bb80457ddd12cf8c29f36d714b5b8f64eefc).

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
