## [Decision Letter]

**Acceptance summary:**

This paper addresses the relationship between the electrophysiological and the anatomical connectomes, utilising a method to describe avalanches of activity. In these rapid events brain activity cascades across cortex. The current paper shows that these avalanches follow the routes of anatomical connections. The result also implies more spatial precision that most would assume possible, which makes the manuscript particularly interesting to M/EEG researchers. The reviewers therefore agree that the paper has broad interest.

**Decision letter after peer review:**

Thank you for submitting your article "The structural connectome constrains fast brain dynamics" for consideration by *eLife*. Your article has been reviewed by 3 peer reviewers, and the evaluation has been overseen by Diego Vidaurre as a Reviewing Editor and Timothy Behrens as the Senior Editor. The following individuals involved in review of your submission have agreed to reveal their identity: Andrew J Quinn (Reviewer #1); George O'Neill (Reviewer #2).

Essential Revisions:

1. Resolve if the results are fundamentally driven by leakage

2. Determine the influence of the region size and SNR on the results.

3. Clarify the description of the method

4. Frame the work into the current literature

*Reviewer #1 (Recommendations for the authors):*

1) The following matlab code illustrates my concern with volume conduction (written in version R2019a). This generates a simple aperiodic signal and linearly weights it into 20 regions before computing an approximation of the avalanches analysis.

%%

rng(42);

a = poly(0.98);

x = filtfilt(1,a,randn(5000,1));

x = x(500:4500); % trim filter edges

% Add two larger pertubations

x(1000:2000) = x(1000:2000) + 400*sin(2*pi*linspace(0,0.5,1001)');

x(2500:3000) = x(2500:3000) + 1000*sin(2*pi*linspace(0,0.5,501)');

% Create + apply linear weights vector

weights = [linspace(0,1,10), linspace(1,0,10)].^3;

y = x * weights;

% Add noise, z-transform and smooth

y = y + randn(4001,1)*60;

y = zscore(y);

y = movmean(y,41,1);

% Illustrate

threshold = 1; % bit low but for illustration

figure;

subplot(211);hold on

plot(y);

plot([0 4001],[threshold threshold],'k:')

plot([0 4001],[-threshold -threshold],'k:')

axis('tight')

title('Linearly mixed signals')

subplot(212)

imagesc(abs(y)'>threshold)

colormap('bone')

title('Avalanches');

%%

Note that in panel 2 on the bottom the onset of the 'active' periods is temporally lagged across regions – sometimes by up to 10s or 100s of samples. The combination of amplitude weighting, noise and z-transforming can therefore induce apparent time-lagged interactions. I don't know how much of the main effect in the paper is driven by this effect but it absolutely must be explored and accounted for. It is not sufficient to say that "The introduction of a time-lag makes it unlikely that our results can be explained trivially by volume conduction"

2) A lot of discussion about the results uses quite forceful causal language.

line 25 "We find that the structural connectome profoundly shapes rapid spreading of neuronal avalanches"

line 100 "We show that the spatial unfolding of neural dynamics at the millisecond scale is shaped by the network of large-scale axonal projections comprising the connectome, thereby constraining exploration of the brain's putative functional repertoire."

Whilst I agree that a structure->function relationship is perhaps the more likely interpretation, this is a correlational study which does not assess whether structure shapes function or vice versa – simply that there is an association. The writing would be improved by acknowledging this and adopting a more cautious interpretation.

3) Some parts of the methods writing is unclear, perhaps as the whole manuscript is so short. For example, I do not understand the second method for estimating transition probabilities outlined here:

"After the initial time-bin of an avalanche, we kept track of what other regions were recruited after the first perturbation. Importantly, we did not scroll through the avalanche in time, as previously described, so as to include time delays as long as the avalanche itself."

4) I am unsure how to interpret the finding that the results are almost completely unchanged by band-pass filtering the data – α, β and γ even have identical R-values. On one hand this could indicate that the results rest completely on wide broadband effects but given the large and well established functional and topographical differences between these oscillations it seems likely that we would expect at least some difference. Particularly as similar past papers do show different structure-function relationships between different MEG frequency bands (https://www.sciencedirect.com/science/article/abs/pii/S1053811918320603). It would be useful for the authors to discuss this point and its potential meaning in more detail.

5) It would be good to see full use of the large datasets presented here. For instance, do the results have test-retest reliability across the two scans analysed per participant? Why only reproduce the finding using the HCP DWI data but not the HCP MEG data?

6) The data appears to be available from the authors 'upon request and conditional on ethics approval' but the analysis code does not appear to be available online.

*Reviewer #2 (Recommendations for the authors):*

I think you are definitely onto something here, but I am not quite sure if this is necessarily the full picture – yet. It's not too difficult to remedy I imagine, but I think on a technical level I'd like to query two main points on this.

I realise that this is a short communication, but it does appear to assume that no work in the M/EEG field on relating structure and function exists, any previous work been done using fMRI. I think it might be important to just give a bit more context where in the literature this might sit? Just looking around for a couple of quick examples of relating white matter tractography data to functional connectivity, there's a study relating EEG connectivity to structural connectivity (Glomb et al., 2020, Network Neurosci. 4 (3): 761-787) And plenty of modelling papers which trying to look at how structure would generate observable M/EEG connectomes (eg. Tewarie et al., 2019 NeuroImage 186). I think a slightly wider look at the literature on this would be really helpful. Crucially, it's not that I don't think that this isn't novel, but rather it seems to currently seems to not acknowledge the exitance of previous work in the M/EEG field?

The use of neuronal avalanches method, is it as invariant to leakage as you initially think it is? Leakage is a product of signal variance (c.f. O'Neill et al., 2015 Phys. Med. Biol R217). So if a signal is stronger in amplitude, the leakage increases and can propagate further. So in principle, if a seed area reaches over a threshold z-score, it could continue to raise in amplitude such that a test area might also then reach a threshold? Could other (i guess more established) connectivity metrics be able to resolve this also? I was wondering even if you assume stationary variance in your signal, you might be able to analytically estimate 'leakage' between two areas (see the O'neill paper above for an analytic expression). Could you then (potentially) have a leakage connectome and see how well that predicts your avalanches perhaps? If its considerably less than the structural, I'd be a lot more confident in what I am seeing.

*Reviewer #3 (Recommendations for the authors):*

Is the in-house software written in Interactive Data Language (used for calculating the number of streamlines connecting each pair of ROIs and the corresponding mean tract length) publicly available? If so, please provide a web link to it.

A bit of extra information about the MEG scanner would also be useful, in addition to reporting the actual number of channels that were manually removed.

[Editors' note: further revisions were suggested prior to acceptance, as described below.]

Thank you for resubmitting your work entitled "The structural connectome constrains fast brain dynamics" for further consideration by *eLife*. Your revised article has been evaluated by Timothy Behrens (Senior Editor) and Diego Vidaurre (Reviewing Editor).

The manuscript has been improved but there are some remaining issues that need to be addressed, as outlined below:

As you can see, all Reviewers are satisfied with the responses, but Reviewer #1 requests some further clarifications on the simulations, and a possible improvement. Since this seems very relevant to the contents of the manuscript, we would like to ask you to go through one more revision.

*Reviewer #1 (Recommendations for the authors):*

Many thanks to the authors for a clear and thorough response to my queries. Overall, the revised manuscript is greatly improved and the changes largely address my concerns.

I must ask for one final clarification, and possible addition, to the simulation scheme looking at leakage. The revision doesn't state whether any noise is added at the sensor level between projection through the leadfields and subsequent beamforming – is any noise added here? If so, please in include any details (and my apologies if I missed this detail somewhere)

If not, then this additional step should be added to the pipeline. As it stands, the simulation without sensor noise creates unusually favourable conditions for the beamformer. Environmental and sensor noise have their own dynamics and can be correlated in time and across sensors, these correlations mask neuronal signals and can be tricky for the beamformer to completely remove.

A gold standard would be to add sensor noise from an empty room recording, or if this is impractical then some modestly scaled noise whose sensor x sensor covariance matrix an example empty room recording.

The correlations of both the 'coupled' and 'uncoupled' simulations are substantially lower than the observed value in the 'correlation of lineally mixed surrogates' figure. I suspect this is partially due to favourable beamforming from missing sensor noise, as it stands the beamformer is able to remove nearly all spurious events. Including the sensor noise would likely induce some more spurious connections in both the coupled and uncoupled cases, it would be a very strong validation of the method to demonstrate that these are still distinguishable from the observed effects.

*Reviewer #2 (Recommendations for the authors):*

The authors have met my initial concerns with the submission. Assuming other reviewers feel the same I am happy with this being published.

*Reviewer #3 (Recommendations for the authors):*

Thank you very much for your revisions. My comments have been addressed satisfactorily.

---

## [Author Response]

Essential Revisions:1. Resolve if the results are fundamentally driven by leakage2. Determine the influence of the region size and SNR on the results.3. Clarify the description of the method4. Frame the work into the current literature

To explore the effects of leakage, we have used a model very similar to the one proposed by reviewer one, and used the leadfield matrices that were used in our study to linearly mix the in-silico data. We explore how this independent but linearly-mixed time series relate to structural connectivity, and we provide evidence that linear mixing alone is unlikely to explain our results. These analyses are shown in the response to reviewer one.

To address the effect of region size, we have normalized the streamline counts by the volume of the regions and re-computed the correlation between the MEG derived transition matrices and the structural connectivity matrices. We find that for both the DKT and the AAL parcellations, the correlations improve after this normalization step. The improvement is more marked for the DKT atlas, which now reaches levels of correlation similar to the ones obtained by the AAL atlas. The analyses reported in the revised manuscript refer to these updated findings, and previous results for the correlations with the non-normalized streamline counts are no longer reported. These analyses are reported in the response to reviewer three.

To address the issue of the potential effect of SNR, we have repeated the analysis including only cortical regions for both the DKT and the AAL. This is done under the hypothesis that, by eliminating deep sources, one takes into account regions that have a comparable SNRs. Further analyses addressing this point are reported in the response to reviewer three.

Finally, we have now framed our manuscript within the current literature, trying to highlight similarities and differences in the interpretation of the data as compared to other similar studies. These changes are mainly reported in the response to reviewer two. In what follows, one can find a detailed, point-to-point response to the reviewers.

Reviewer #1 (Recommendations for the authors):1) The following matlab code illustrates my concern with volume conduction (written in version R2019a). This generates a simple aperiodic signal and linearly weights it into 20 regions before computing an approximation of the avalanches analysis.%%rng(42);a = poly(0.98);x = filtfilt(1,a,randn(5000,1));x = x(500:4500); % trim filter edges% Add two larger pertubationsx(1000:2000) = x(1000:2000) + 400*sin(2*pi*linspace(0,0.5,1001)');x(2500:3000) = x(2500:3000) + 1000*sin(2*pi*linspace(0,0.5,501)');% Create + apply linear weights vectorweights = [linspace(0,1,10), linspace(1,0,10)].^3;y = x * weights;% Add noise, z-transform and smoothy = y + randn(4001,1)*60;y = zscore(y);y = movmean(y,41,1);% Illustratethreshold = 1; % bit low but for illustrationfigure;subplot(211);hold onplot(y);plot([0 4001],[threshold threshold],'k:')plot([0 4001],[-threshold -threshold],'k:')axis('tight')title('Linearly mixed signals')subplot(212)imagesc(abs(y)'>threshold)colormap('bone')title('Avalanches');%%Note that in panel 2 on the bottom the onset of the 'active' periods is temporally lagged across regions – sometimes by up to 10s or 100s of samples. The combination of amplitude weighting, noise and z-transforming can therefore induce apparent time-lagged interactions. I don't know how much of the main effect in the paper is driven by this effect but it absolutely must be explored and accounted for. It is not sufficient to say that "The introduction of a time-lag makes it unlikely that our results can be explained trivially by volume conduction"

We agree on the importance of this issue. We explored this point to check if spatial leakage could explain the correlation we found between the transition and structural connectivity matrices.

Firstly, on the same lines suggested by referee one, we simulated n simple aperiodic signals, with n = number of regions (sources) = 66, since we focused on cortical regions. Consistent with the referee’s suggestions, we generated n white Gaussian processes and introduced a temporal correlation in each, using a zero-phase, all-pole polynomial filter. Then, for 100 distinct times, we added a large perturbation to randomly selected regions, with a duration selected randomly among 5, 10 or 100 samples, and with an amplitude randomly selected between 50 and 400, as to explore the effects of perturbations of different intensities. We ensured that at least 200 samples separate any two subsequent perturbations in time. In other words, two perturbations cannot overlap in this surrogate data. However, the same region might have more than one perturbation over time. As such, we refer this surrogate data as “uncoupled”. We repeated this procedure for each subject, such that each of the 47 participants had a set of “uncoupled”, time series, each containing 100, non-overlapping random perturbations generated at random locations. The “uncoupled” dataset represents local perturbations that occur at random without any influence from brain structure.

Based on the “uncoupled” surrogate data, we simulated, for each subject, time-series coupled by the subject-specific structural connections inferred from tractography. To do this, we modified the uncoupled time-series as follows: the activity of each region *i* at time t was given by the summation of the activity of every other region *j*, at time t – δ, weighted by the normalized streamline count between regions *i* and , as in :icoupled(t)=iuncoupled(t)+∑j=1i≠jj(t−δ).Wijwhere i_coupled_(t) is the value of the new coupled time series at time t, *i*_uncoupled_*(t)* is the corresponding value of the uncoupled i^th^ series at time t, N is the number of regions, *j(t-δ)* is the value of the j^th^ uncoupled time series at time t – δ, with δ = 15 samples, W*_ij_* is the streamline count of the ij^th^ structural edge.

We named this set of surrogate data “coupled”. While this only phenomenologically simulates a first-order dependence with homogeneous delays, it suffices to make a point about field spread, and it is not intended (of course) to simulate any mechanism.

We applied the same statistical procedures described in our original manuscript to the surrogate data (uncoupled and coupled). In particular, we z-scored the surrogate data, applied a threshold (z>±3), computed neuronal avalanches and the corresponding transition matrix (including all avalanche sizes). Finally, as per our original analysis, we averaged transition probabilities across individuals and then symmetrized the obtained transition matrix. In Author response image 1, one can see the resulting transition matrices for the uncoupled and coupled surrogate data, as well as the structural connectivity matrix to the right.

We correlated, using Spearman’s correlation, each of the transition matrices (coupled and uncoupled) with the structural connectivity matrix. As expected, in the case of the uncoupled regions, the corresponding TM does not relate to structural connectivity. However, the TM corresponding to the coupled time series, showed a direct correlation with the dti (r=0.33, p<0.0001).Based on these surrogate models, we went on to study the effect of leakage. To this end, we considered the subject-specific lead-field matrices, that had been used to beamform the original data. Hence, each one of the 47 subject-specific leadfields (i.e. one per each of the 47 participants) was used to project the surrogate data to the sensor space, obtaining 47 sets of time-series in the sensor space, derived from the uncoupled data, and 47 sensor space time series based on the coupled data. Note that the leadfield estimates the signal at the location of each sensor as a linear combination of all the sources. Finally, we estimated the beamformer weights from this newly generated sensor-space time-series, and reconstructed new source-level time-series. This approach enables the application of the mixing matrix (leadfield) that was actually used per each subject to compute the data obtained in the main results. As such, so we believe this allows us to be as adherent as possible to the procedure that was actually used to estimate the main results presented in the manuscript.

Using these newly reconstructed source-level time-series we applied the same statistical procedures to map transition matrices for the surrogate datasets, , which are shown, also in log scale, in Author response image 2.

**Author response image 2. respfig2:** 

Similarly as before, we correlated the transition matrix with the structural connectivity matrix using Spearman’s correlation. As expected, one can see that a correlation exists between the coupled surrogate and the structural matrix (r=0.12, p=0.001), but not for the uncoupled surrogate (r=0.02, p=0.59). The absence of an association between transition and structural connectivity in the uncoupled surrogate data suggests that the potential effects of leakage are an unlikely explanation for the associations reported in the real data. Importantly, the relationship that we observed in real data is stronger than the one we obtained from the simulated coupled data. Multiple reasons could explain this fact, including that our coupled model is limited to first-order dependencies in structure, while multiple lines of evidence show that white matter bundles also induce dependencies of higher order (Seguin et al., 2020).The entire procedure described above was carried out one hundred times, whereby a new random sequence (generated as described above) per each participant was generated, multiplied by the subject-specific leadfield matrix, source-reconstructed, then the subject-specific transition matrices were computed, and averaged per group. The distribution of the retrieved 100 correlations is reported in Author response image 3 for the case of coupled surrogates:

**Author response image 3. respfig3:** 

It is easy to see that the randomly generated sequences of uncoupled perturbations, which are then mixed linearly exactly as they real data were mixed, do not reproduce high correlations with structural connectivity, hence making it unlikely that linear mixing alone is a plausible explanation of the relationship that we found.This analysis has been partly reported as a new section in the methods, as follows:

“Field spread analysis

Volume conduction alone is an unlikely explanation of our results, given that simultaneous activations do not contribute to the transition matrix, due to the time lags introduced. […] We repeated the entire procedure reported above one hundred times, and show that is unlikely that linear mixing alone can explain the significant association between transition probabilities and structural connectivity (p<0.001).”

Finally, a small note on the code that reviewer 1 sent us. Firstly, we wish to acknowledge the high quality of the review, and the significant efforts of the reviewer. We are sincerely grateful. We suggest that a small modification to the code would better align this toy example with the procedure used in the manuscript. Namely, we would change y = x * weights to y =(x + randn(4001,1)*60) * weights; and eliminate the following line: y = y + randn(4001,1)*60. This would better conform to our case since noise also should be scaled by the weights. When running the code with such modifications, the output figure becomes – see Author response image 4:

**Author response image 4. respfig4:** 

In other words, the z-score has now normalized all signals equally, as in the procedure we used. The constant turquoise line at y=0 is due to one weight that was set to zero. We understand that these are toy models, and we are only hoping to illustrate how subtle changes to the model can lead to significant changes. However, the shape of the avalanches and some of the statistical properties that appear phenomenologically could not be explained easily by such a toy model (in our opinion, shape collapses are particularly convincing to this regard – please refer to the answer to referee three as well for more considerations and analyses on this point).

2) A lot of discussion about the results uses quite forceful causal language.line 25 "We find that the structural connectome profoundly shapes rapid spreading of neuronal avalanches"line 100 "We show that the spatial unfolding of neural dynamics at the millisecond scale is shaped by the network of large-scale axonal projections comprising the connectome, thereby constraining exploration of the brain's putative functional repertoire."Whilst I agree that a structure->function relationship is perhaps the more likely interpretation, this is a correlational study which does not assess whether structure shapes function or vice versa – simply that there is an association. The writing would be improved by acknowledging this and adopting a more cautious interpretation.

The sentence on line 25 was changed to: “We find that the structural connectome relates to, and likely affects, the rapid spreading of neuronal avalanches”.

The sentence on line 100 was changed to: “We show that the spatial unfolding of neural dynamics at the millisecond scale relates to the network of large-scale axonal projections comprising the connectome, likely constraining the exploration of the brain’s putative functional repertoire.”

The language in the sentences has been adjusted as suggested. We have aimed to avoid jargon and keep the language as simple as possible to ensure that the paper is accessible to clinicians and neuroscientists. We believe that this framework might allow clinical researchers to test the effects of localized lesions on the large-scale dynamics and, perhaps, on symptoms.

3) Some parts of the methods writing is unclear, perhaps as the whole manuscript is so short. For example, I do not understand the second method for estimating transition probabilities outlined here:"After the initial time-bin of an avalanche, we kept track of what other regions were recruited after the first perturbation. Importantly, we did not scroll through the avalanche in time, as previously described, so as to include time delays as long as the avalanche itself."

We apologize, and agree that the explanation was not clear. We have now added a more through explanation that reads:

“In short, we identified regions that were recruited in an avalanche after the first perturbation (i.e. the initial time-bin of an avalanche). Since we did not scroll through the avalanche in time, as previously described, we considered time delays as long as the avalanche itself, while minimizing the influence of short delays. This means that the avalanche-specific transition matrix is now binary, and the *ij^th^* element is equal to 1 if region *i* started the avalanche (i.e. it was active at the first time-bin) and region *j* was recruited in the avalanche at any subsequent timepoint, and 0 otherwise.”

4) I am unsure how to interpret the finding that the results are almost completely unchanged by band-pass filtering the data – α, β and γ even have identical R-values. On one hand this could indicate that the results rest completely on wide broadband effects but given the large and well established functional and topographical differences between these oscillations it seems likely that we would expect at least some difference. Particularly as similar past papers do show different structure-function relationships between different MEG frequency bands (https://www.sciencedirect.com/science/article/abs/pii/S1053811918320603). It would be useful for the authors to discuss this point and its potential meaning in more detail.

We agree with this point. Our results show that a correlation between perturbations and structural scaffolding exists in all frequency bands. This is only meant, again, as a demonstration that structure affects the large scale organization of the brain at many levels, and it is not only limited to specific frequencies, as shown earlier. However, the avalanches are distinct across frequency bands. For example, avalanches do not occur (necessarily) simultaneously in the different frequency bands.

We have added and briefly discussed the reference that the referee indicated. That is indeed very relevant, thank you. However, as the reviewer noted, our results are solely based on the aperiodic part of activity. In a sense, one could say that the part of the signal we used is the one that Tewarie et al. did not consider (by focusing on the AEC and, more generally, making assumptions of stationarity), and vice versa. How aperiodic, scale-free activity reconciliates with coordination that relates to steady oscillations and synchrony remains to be elucidated. We hope that this evidence can help to provide observables related to this, against which theoretical predictions could be tested. We have integrated this within a broader review of the current literature included in the discussion, as follows:

“While our findings were replicated across multiple frequency bands, structural connectivity can potentially impose frequency-dependent constraints on avalanche spread. Future work should investigate frequency-specific data to understand what leads to the emergence of avalanches and, most importantly, to the specific spatio-temporal patterns of recruited regions that defines individual (or at least groups of) avalanches in each specific frequency-band.”.

Please also consider the response to question one of reviewer two, where further addition to the discussion also addressed the point raised here (and cite the literature suggested by the reviewers).

5) It would be good to see full use of the large datasets presented here. For instance, do the results have test-retest reliability across the two scans analysed per participant? Why only reproduce the finding using the HCP DWI data but not the HCP MEG data?

For the DKT analysis, we have checked the correlation, both at the individual and at the group level, for the two scans separately. The results are reported in Author response image 5:

**Author response image 5. respfig5:** 

Interestingly, when both halves are considered separately, the correlation between transition matrices and structural connectivity is comparable, providing evidence of robustness of the data. However, in both analyses, it is somewhat lower than the one we observed when taking both halves into account. Hence, we suggest this is evidence that, if the time-series are too short, some of the structure of the data might become difficult to measure and, accordingly, more data allows further convergence due to improved SNR. With regard to the HCP MEG data, we chose to replicate the results using the HCP tractography estimates so as to rule out that tractography performed on a 1.5 tesla machine might have introduced biases invalidating the results. Hence, this analysis is not entirely a replication of the results, but was done to rule out a specific concern. On the other hand, a subject-wise replication of the tractography and MEG correlation is a replication, which is desirable, but not strictly hindering the validity of the results of the current study. We have added a short sentence mentioning this, that reads: “However, further studies in independent datasets will have to replicate the current results”.

6) The data appears to be available from the authors 'upon request and conditional on ethics approval' but the analysis code does not appear to be available online.

We apologize for this. Actually, the code for the main analysis is available on GitHub, at https://github.com/pierpaolosorrentino/Transition-Matrices-. This was stated in the submission process, we thought the reviewer would have access to this information. We apologize again for the inconvenience.

Reviewer #2 (Recommendations for the authors):I think you are definitely onto something here, but I am not quite sure if this is necessarily the full picture – yet. It's not too difficult to remedy I imagine, but I think on a technical level I'd like to query two main points on this.I realise that this is a short communication, but it does appear to assume that no work in the M/EEG field on relating structure and function exists, any previous work been done using fMRI. I think it might be important to just give a bit more context where in the literature this might sit? Just looking around for a couple of quick examples of relating white matter tractography data to functional connectivity, there's a study relating EEG connectivity to structural connectivity (Glomb et al., 2020, Network Neurosci. 4 (3): 761-787) And plenty of modelling papers which trying to look at how structure would generate observable M/EEG connectomes (eg. Tewarie et al., 2019 NeuroImage 186). I think a slightly wider look at the literature on this would be really helpful. Crucially, it's not that I don't think that this isn't novel, but rather it seems to currently seems to not acknowledge the exitance of previous work in the M/EEG field?

We apologize if we gave the impression of not acknowledging previous work. This was not our intention. We chose to aim at a brief report and not a full paper since we would like to convey our novel findings as succinctly as possible. We have now cited some of the work indicated by the reviewer, and, in the discussion, we now frame our work in more detail within the current literature, as follows:

“Consistent with our findings, two recent M/EEG studies showed that functional connectivity, as estimated using amplitude-envelope coupling (AEC), relates to structural connectivity (Glomb et al., 2020; Tewarie et al., 2019). However, in contrast to AEC, we conducted time-resolved analyses, characterizing avalanche dynamics at high temporal resolution. Further work is needed to determine the extent to which structure-function coupling is dynamic. To this regard, our results suggest that coupling is strongest during avalanche events, consistent with established theories (Dehaene et al., 1998)”.

The use of neuronal avalanches method, is it as invariant to leakage as you initially think it is? Leakage is a product of signal variance (c.f. O'Neill et al., 2015 Phys. Med. Biol R217). So if a signal is stronger in amplitude, the leakage increases and can propagate further. So in principle, if a seed area reaches over a threshold z-score, it could continue to raise in amplitude such that a test area might also then reach a threshold? Could other (i guess more established) connectivity metrics be able to resolve this also? I was wondering even if you assume stationary variance in your signal, you might be able to analytically estimate 'leakage' between two areas (see the O'neill paper above for an analytic expression). Could you then (potentially) have a leakage connectome and see how well that predicts your avalanches perhaps? If its considerably less than the structural, I'd be a lot more confident in what I am seeing.

We thank the reviewer for this relevant comment. Please refer to our response to reviewer one (comment 1) for additional analysis exploring the effects of leakage. We chose not to pursue the idea of using more traditional metrics (plv, pli, plm etc.) since they are based on the framework of synchronization and on the assumption of stationarity. In contrast, our analysis focuses solely on the aperiodic, fat tailed distributed part of the activity, which is what is not to be expected in a purely oscillatory system. A comment on this was added to the manuscript, as part of the broader review of the literature in the discussion.

Reviewer #3 (Recommendations for the authors):Is the in-house software written in Interactive Data Language (used for calculating the number of streamlines connecting each pair of ROIs and the corresponding mean tract length) publicly available? If so, please provide a web link to it.

The software is not publicly available. However, it can be provided for research use upon request to the author [MQ]

A bit of extra information about the MEG scanner would also be useful, in addition to reporting the actual number of channels that were manually removed.

Further information about the MEG system has been provided, as well as a reference that fully describes the system we used, as well as the number of channels that were manually removed:

“The MEG system we used was equipped with 163 magnetometers, and was developed by the National Research Council of Italy at the Institute of Applied Sciences and Intelligent Systems (ISASI). All technical details regarding the MEG device are reported in (Rombetto et al., 2014).”

And

“(136 ± 4 sensors were kept)”

References

Abeyasinghe PM, Aiello M, Cavaliere C, Owen AM, Soddu A. 2021. A comparison of diffusion tractography techniques in simulating the generalized Ising model to predict the intrinsic activity of the brain. *Brain Struct Funct* 226:817–832. doi:10.1007/s00429-020-02211-6

Dehaene S, Kerszberg M, Changeux J-P. 1998. A neuronal model of a global workspace in effortful cognitive tasks. *PNAS* 95:14529–14534. doi:10.1073/pnas.95.24.14529

Glomb K, Mullier E, Carboni M, Rubega M, Iannotti G, Tourbier S, Seeber M, Vulliemoz S, Hagmann P. 2020. Using structural connectivity to augment community structure in EEG functional connectivity. *Network Neuroscience* 4:761–787. doi:10.1162/netn_a_00147

Honey CJ, Sporns O, Cammoun L, Gigandet X, Thiran JP, Meuli R, Hagmann P. 2009. Predicting human resting-state functional connectivity from structural connectivity. *Proceedings of the National Academy of Sciences of the United States of America* 106:2035–2040. doi:10.1073/pnas.0811168106

Rombetto S, Granata C, Vettoliere A, Russo M. 2014. Multichannel System Based on a High Sensitivity Superconductive Sensor for Magnetoencephalography. *Sensors* 14:12114–12126. doi:10.3390/s140712114

Seguin C, Razi A, Zalesky A. 2019. Inferring neural signalling directionality from undirected structural connectomes. *Nature Communications* 10:1–13. doi:10.1038/s41467-019-12201-w

Seguin C, Tian Y, Zalesky A. 2020. Network communication models improve the behavioral and functional predictive utility of the human structural connectome. *Network Neuroscience* 4:980–1006. doi:10.1162/netn_a_00161

Seguin C, Van Den Heuvel MP, Zalesky A. 2018. Navigation of brain networks. *Proceedings of the National Academy of Sciences of the United States of America* 115:6297–6302. doi:10.1073/pnas.1801351115

Shriki O, Alstott J, Carver F, Holroyd T, Henson RNA, Smith ML, Coppola R, Bullmore E, Plenz D. 2013. Neuronal avalanches in the resting MEG of the human brain. *Journal of Neuroscience* 33:7079–7090. doi:10.1523/JNEUROSCI.4286-12.2013

Tewarie P, Abeysuriya R, Byrne Á, O’Neill GC, Sotiropoulos SN, Brookes MJ, Coombes S. 2019. How do spatially distinct frequency specific MEG networks emerge from one underlying structural connectome? The role of the structural eigenmodes. *Neuroimage* 186:211–220. doi:10.1016/j.neuroimage.2018.10.079

[Editors' note: further revisions were suggested prior to acceptance, as described below.]

Reviewer #1 (Recommendations for the authors):Many thanks to the authors for a clear and thorough response to my queries. Overall, the revised manuscript is greatly improved and the changes largely address my concerns.I must ask for one final clarification, and possible addition, to the simulation scheme looking at leakage. The revision doesn't state whether any noise is added at the sensor level between projection through the leadfields and subsequent beamforming – is any noise added here? If so, please in include any details (and my apologies if I missed this detail somewhere)If not, then this additional step should be added to the pipeline. As it stands, the simulation without sensor noise creates unusually favourable conditions for the beamformer. Environmental and sensor noise have their own dynamics and can be correlated in time and across sensors, these correlations mask neuronal signals and can be tricky for the beamformer to completely remove.A gold standard would be to add sensor noise from an empty room recording, or if this is impractical then some modestly scaled noise whose sensor x sensor covariance matrix an example empty room recording.The correlations of both the 'coupled' and 'uncoupled' simulations are substantially lower than the observed value in the 'correlation of lineally mixed surrogates' figure. I suspect this is partially due to favourable beamforming from missing sensor noise, as it stands the beamformer is able to remove nearly all spurious events. Including the sensor noise would likely induce some more spurious connections in both the coupled and uncoupled cases, it would be a very strong validation of the method to demonstrate that these are still distinguishable from the observed effects.

We thank the reviewer for their important suggestion. We have now repeated the same simulation as earlier, but this time adding white noise, correlated as 1/*distanceamong sensors*, to the sensor-level signal, before performing the inversion. Each channel has SNR = 4. We report in Author response image 6, to the right, the correlations retrieved with the added noise, and to the left the correlations obtained without the noise, from Author response image 3, for comparison.

**Author response image 6. respfig6:** 

As correctly predicted by reviewer one, the noise structure introduces spurious correlations, that can be observed in both the coupled and uncoupled time-series, given the shift of both distributions to the right, as compared to the correlations retrieved without noise. However, given that in both cases the observed correlations are significantly lower than the original ones is reassuring. We have now modified the manuscript, in the methods, field spread analysis section, as follows:“Noise, correlated as 1/*distance among sensors*, was then added to the sensor-level time series, with a SNR = 4.”

We have also replaced our original simulation results (no noise) with the new results that include noise.